# Chitosan Production by Fungi: Current State of Knowledge, Future Opportunities and Constraints

Silvia Crognale, Cristina Russo [ID], Maurizio Petruccioli [ID] and Alessandro D'Annibale *[ID]

Department for Innovation in Biological, Agro-Food and Forest Systems (DIBAF), University of Tuscia, Via S. Camillo De Lellis snc, 01100 Viterbo, Italy; crognale@unitus.it (S.C.); cristina.russo@hotmail.it (C.R.); petrucci@unitus.it (M.P.)

\* Correspondence: dannib@unitus.it

**Abstract:** Conventionally, the commercial supply of chitin and chitosan relies on shellfish wastes as the extraction sources. However, the fungal sources constitute a valuable option, especially for biomedical and pharmaceutical applications, due to the batch-to-batch unsteady properties of chitin and chitosan from conventional ones. Fungal production of these glycans is not affected by seasonality enables accurate process control and, consequently, more uniform properties of the obtained product. Moreover, liquid and solid production media often are derived from wastes, thus enabling the application of circular economy criteria and improving the process economics. The present review deals with fungal chitosan production processes focusing on waste-oriented and integrated production processes. In doing so, contrary to other reviews that used a genus-specific approach for organizing the available information, the present one bases the discussion on the bioprocess typology. Finally, the main process parameters affecting chitosan production and their interactions are critically discussed.

**Keywords:** chitosan; fermentation; waste upgrading; integrated bioprocesses; fungi

## 1. Introduction

Chitin, a structural glycan composed of randomly distributed *N*-acetyl-D-glucosamine (GlcNAc) residues (Figure 1a) [1], is the second most abundant biopolymer on earth (more than 100 billion tons) [2]. Chitosan, a linear heteroglycan mainly made of β-(1-4)-linked D-glucosamine (GlcN) units (Figure 1b), is often derived from chitin deacetylation, the extent of which, however, is never quantitative. Consequently, chitosan is a copolymer made of GlcNAc and GlcN residues, where the latter account for at least 60% of total residues [1,3,4]. However, the degree of deacetylation (DD) of commercially available chitosans generally amounts to or exceeds 80% [3].

Over a 2020–2027 period, the Global Industry Analysts Inc. [5] estimated growth of the market of chitin and chitosan from $106.9 \times 10^3$ to around $282 \times 10^3$ tons, with a compounded annual growth rate equal to 14.8%; this growth estimate was attributed to the ever-increasing applications of these polymers in various end-use sectors.

Chitin occurs in nature in different crystalline forms denominated α-, β-, and γ-chitin, exhibiting distinct physicochemical properties (Figure 1c). The differences among these polymorphs are due to the mode with which crystalline regions' chains are reciprocally arranged. In the α and β forms, all the chains are arranged in an antiparallel and parallel mode, respectively, while in the γ form, there is an alternation of sets of two parallel strands with single antiparallel ones [6]. Among these allomorphs, α-chitin is the most widespread being found in arthropods and fungi; β-chitin generally occurs in cephalopods, while γ-chitin is rather rare.

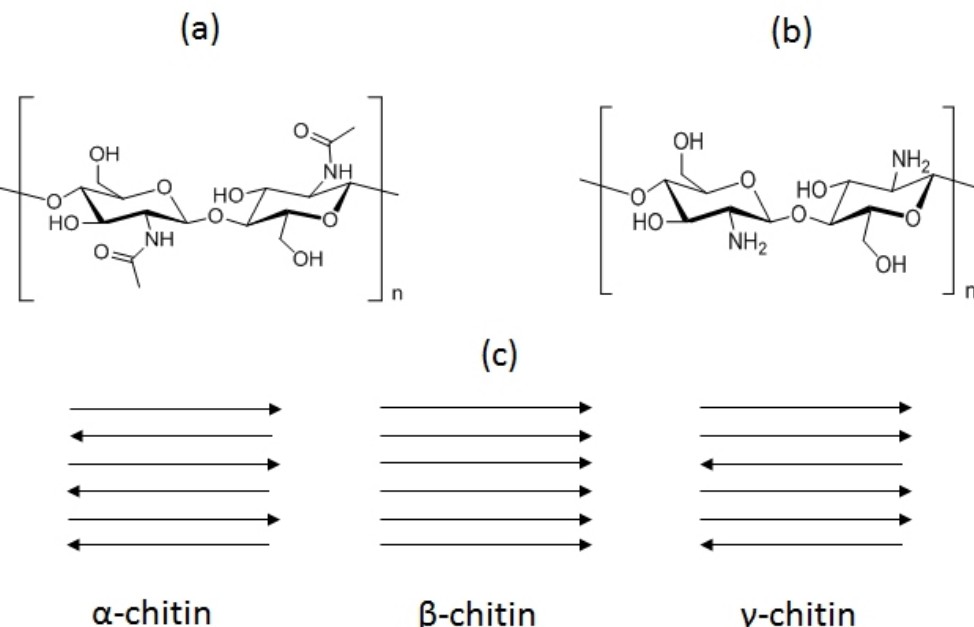

**Figure 1.** Basic structures of chitin (**a**) and chitosan (**b**) and chitin allomorphs (**c**). The tips of the arrows indicate the positions of the reducing ends of the chains.

At present, the industrial production of chitin and chitosan relies on crustacean wastes. Chitin contents in crab and shrimp processing wastes range from 13% to 15% and 14% to 27% dry weight, respectively [7]. The heterogeneous composition of crustacean shell wastes requires stepwise chemical methods to extract chitin and chitosan from these sources [8].

The extraction scheme generally involves a demineralization step with strong acids, followed by alkaline deproteinization and, frequently, a decolorization step. The first step, generally relying on HCl solutions (concentration ranges 0.6–11.0 M), is generally conducted at room temperatures, while the second uses NaOH solutions (concentration ranges 0.12–5.0 M) and temperature up to 160 °C; the final step, frequently added to remove pigments, such as β-carotene and astaxanthin, generally relies on acetone as the extraction solvent [9]. The chitin thus obtained undergoes alkaline deacetylation under very harsh chemical conditions (NaOH solutions from 30% to 60%) under variable temperature and contact times to yield chitosan [10]. The variability of the raw materials and the harsh conditions that characterize some extraction steps can lead to unsteady physicochemical properties of chitosan from batch to batch [9].

The occurrence of chitin and chitosan in fungi has opened the door to a promising alternative route for their productions [11]. Since fungal chitin has a lower ash content than crustacean shell wastes, the demineralization step is not required during its processing [12,13]. Moreover, chitin and chitosan of fungal origin provide a non-seasonal and reliable source of these polymers and consistent properties of the product. The extraction of a value-added product, such as chitosan, may afford a profitable solution to mushroom growers and biotechnological industries considering the vast quantities of fungal-based wastes accumulated and the ensuing expense in waste management.

## 2. Physicochemical and Functional Properties of Chitin and Chitosan

The chemical structures of chitin and chitosan resemble that of cellulose, a glycan composed of hundreds of D-glucose residues connected by β-(1-4) linkages [1]. In chitin and chitosan, however, an acetamide or amino group replaces the hydroxyl group at the C-2 position of glucose residues. Thus, the nitrogen content of chitin and chitosan ranges from 5% to 8%, and the presence of amino groups gives these glycans distinctive biological functions and susceptibility to chemical modification reactions [14]; chitosan, in particular, owing to the presence of free amino groups is susceptible to *N*-acylation and

Schiff's reactions paving the way to a variety of chemical modifications. Moreover, the joint presence of the amino and hydroxyl groups on each deacetylated unit renders chitosan more water-soluble and chemically reactive than chitin. Due to their $pk_a$ values ($\approx$6.3), the free amino groups in GlcN residues are protonated at slightly acid pHs, making chitosan the only naturally occurring cationic glycan [15]. A further consequence is the polymer's solubility in slightly acid aqueous solution as opposed to chitin [16].

The majority of the biological properties of these glycans are due to their physico-chemical features, such as solubility, deacetylation degree, molecular weight, and inherent moisture content [17]. For instance, the inhibition of fungal and bacterial growth exerted by chitosan relies principally on the extent of positively charged groups and molecular mass. Two main mechanisms have been suggested to explain the antimicrobial activity of chitosan. The first outlines the importance of its molecular weight and postulates that the smaller chito-oligosaccharides can easily penetrate the cellular membrane, thus preventing cell growth via inhibition of DNA transcription [18,19]. The second suggests that the positively charged groups of chitosan interact with anionic components of the microbial cell membrane resulting in cell death [20,21]. Moreover, chitosan can operate as a chelator of essential elements [22]. Chitosans with a degree of deacetylation (DD) larger than 97.5% have a higher positive charge density and an ensuing stronger antibacterial activity than those with moderate DD (83.7%), as shown by Kong et al. [23]. Some properties of chitin and chitosan, such as non-toxicity, biodegradability, biocompatibility, and non-allergenicity, associated with bioactivity and suitable adsorption properties, render them appropriate alternative options to artificial polymers [24,25]. Another reason underlying their success is that they can be manufactured to yield several forms, including beads, flakes, membranes, gels, and fibers [15]. As a consequence, they have been exploited as carriers for enzyme immobilization [26,27], coagulating agents in effluents treatment, as food preservatives [28], hypocholesterolemic, and wound healing agents, and as components of several drug delivery systems [29,30].

## 3. General Aspects of Chitin and Chitosan Production from Fungal Sources

Since the beginning of this century, many countries have focused attention on using fungal sources for the commercial production of chitin and chitosan due to the remarkable disadvantages that burden the conventional process. Table 1 comparatively summarizes the advantages and disadvantages of chitin and chitosan production from fungal and crustacean sources [12,13].

Among the advantages of the fungal approach, there is the possibility of obtaining chitosans with different properties by varying species and culture conditions [31,32]. For instance, the chitosan derived from shellfish wastes has a high molecular mass (around $1.5 \times 10^6$ Da), while the MW of chitosan from fungal sources widely ranges from $6.4 \times 10^3$ to $1.4 \times 10^6$ Da [33,34]. High MW chitosans are sparingly soluble in neutral pH aqueous solutions and yield high viscous solutions that limit their exploitation in the food, health, and agricultural sectors [35]. Fungal-derived chitosan with medium-low MW can be used as hypocholesterolemic agents in healthcare products and as a thread or membrane in a variety of biomedical applications [36,37]. Moreover, chitosan extraction from fungal sources is more environmentally benign than that from shellfish wastes since the latter source requires highly concentrated acid and alkaline solutions for demineralization and chemical deacetylation that have to be disposed of. Another advantage associated with fungal chitosan encompasses the absence of allergenic proteins, such as tropomyosin [9]. Chitin and chitosan extraction from fungi can lower disposal costs of fungal-based waste materials in association with the production of value-added products, which may offer a lucrative opportunity to the biotechnological industries [38–40].

### 3.1. Chitin and Chitosan Biosynthesis and Their Biological Functions in Fungi

Arthropods and fungi share a common biosynthetic pathway that uses glucose and its storage carbohydrates, such as trehalose and glycogen, as the starting materials. The

pathway of chitin biosynthesis is organized into three groups of reactions, the first leading to the formation of GlcNAc, the second yielding its activated counterpart uridine 5′-diphospho-*N*-acetylglucosamine (UDP-GlcNAc) through a modification of the Leloir pathway, and the third resulting in polymer formation using UDP-GlcNAc as the GlcNAc donor to the growing chitin chain (Figure 2) [41].

**Table 1.** Benefits and drawbacks of fungal chitosan compared to those from conventional sources with reference to a series of evaluation criteria.

| Evaluation Criterion | Benefits | Drawbacks |
|---|---|---|
| Biomass supply | Not affected by seasonal and geographical factors. Possible biomass supply from pharmaceutical and biotechnological industries. | Lower biomass amounts than those made available from the shellfish industry. |
| Extraction process | Process scheme simpler (no demineralization and decolorization steps) and lower amounts of chemicals employed as compared to those used for crustacean sources. | Less established as compared to that from shellfish waste. |
| Environmental impact and disposal costs of process wastes | More environmentally friendly and lower disposal costs of effluents as compared to the shellfish waste process. | Potential risks of dispersal of pathogenic fungi when dealing with species not satisfying the generally regarded as safe requirements. |
| Inorganic and organic contaminants in the product | Absence of heavy metals and allergenic proteins as opposed to chitosan preparations from shellfish waste. | Some chitosan preparations might contain residual phosphates. |
| Production costs | They can be modulated by the choice of low-cost substrates and low equipment-intensive fermentation techniques, such as solid-state fermentation. | Not yet competitive in terms of production costs compared to the conventional process. |
| Physicochemical properties of the products | Molecular weights and degree of deacetylation of fungal chitosans frequently lower and higher, respectively, than those from conventional sources with ensuing positive impacts on their antimicrobial and antioxidant activities. | The lower MW of fungal chitosans than those from conventional sources make them less suitable as anti-lipidemic and hypocholesterolemic agents. |

Free glucose or that derived from trehalase-catalyzed hydrolysis of trehalose is converted to glucose-6-phosphate (G-6-P) by hexokinase. If the starting material is glycogen, its depolymerization, catalyzed by glycogen phosphorylase, releases glucose-1-phosphate, which is also converted to glucose-6-phosphate by phosphoglucomutase-catalyzed isomerization. Irrespective of its origin, G-6-P is then isomerized to fructose-6-phosphate (F-6-P) by phosphoeosisomerase. F-6-P is then converted to *N*-acetyl-D-glucosamine-6-phosphate (GlcNAc-6-P) through two consecutive transfer reactions of an amino and acetyl group where glutamine and acetyl CoA, respectively, act as the donors. Isomerization step of GlcNAc-6-P catalyzed by phospho-*N*-acetyl glucosamine mutase yields 1-phospho-*N*-acetyl-D-glucosamine (GlcNAc-1-P). The chitin precursor, UDP-GlcNAc, is formed upon reaction of GlNAc-1-P with uridine triphosphate (UTP) and serves as a N-acetylglucosamine (GlNAc) donor for the sequential addition of GlcNAc units to the non-reducing terminus of the growing chain catalyzed by chitin synthase [42]. The linear chains spontaneously assemble to form microfibrils with varying diameters and lengths. In a further step, chitin deacetylase (CDA, E.C. 3.5.1.41) brings about the deacetylation of GlcNAc residues of chitin, thus leading to chitosan [43]. The formation of GlcNAc and UDP-GlcNAc takes place in the cytosol, while chitin synthesis occurs in specialized domains of the cell membrane. Some enzymes involved in chitin syntheses, such as glutamine-fructose-6-phosphate ami-

dotransferase (EC 2.6.1.16), UDP-N-acetylglucosamine pyrophosphorylase (EC 2.7.7.23), and chitin synthase (EC 2.4.1.16), are subjected to tight regulation and limit the rate of chitin production [41,44].

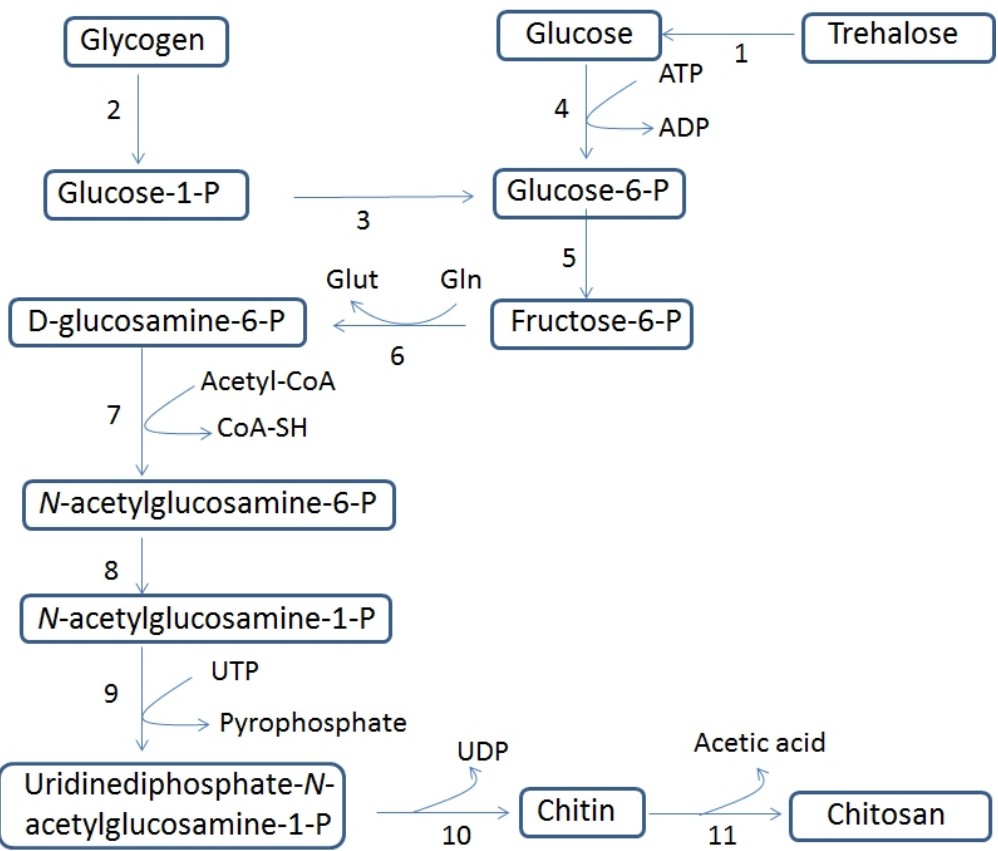

**Figure 2.** Biosynthetic pathway of chitin and chitosan. The following numbering has been assigned to the enzymes that catalyze each reaction: 1, trehalase; 2, glycogen phosphorylase; 3, phosphoglucomutase; 4, hexokinase; 5, glucose-6-phosphate isomerase; 6, glutamine-fructose-6-phosphate amidotransferase; 7, glucosamine-6-phosphate *N*-acetyltransferase; 8, *N*-acetylglucosamine-phosphate mutase; 9, UDP-*N*-acetylglucosamine pyrophosphorylase; 10, chitin synthase and 11, chitin deacetylase. The abbreviations used are as follows: acetyl CoA, acetyl coenzyme A; ADP, adenosine diphosphate; CoASH, coenzyme A; Gln, L-glutamine; Glu, L-glutamate; UDP, uridine diphosphate; UTP, uridine triphosphate.

Table 2 summarizes the taxonomic classification of chitosan-producing species and range of chitosan contents (referred to dry biomass) and degree of deacetylation.

**Table 2.** Taxonomic classification of chitosan-producing species and range of chitosan contents referred to their biomass dry weight and degree of deacetylation (DD).

| Genus and Species | Class | Order | Family | Chitosan Content (g/kg) | DD (%) | References |
|---|---|---|---|---|---|---|
| *Absidia*<br>*A. blakesleeana*<br>*A. coerulea*<br>*A. glauca*<br>*A. orchidis* | Zygomycetes | Mucorales | Cunninghamellaceae | 10–170<br>30–300<br>52–59<br>18–69 | 85<br>93–95<br>75–80<br>68–85 | [31,45–51] |
| *Cunninghamella*<br>*C. bertholletiae*<br>*C. echinulata*<br>*C. elegans*<br>*C. ramose* | Zygomycetes | Mucorales | Cunninghamellaceae | 55–128<br>50–130<br>35–78<br>50–123 | 87–90<br>85<br>72–90<br>n.r. | [27,51–59] |
| *Gongronella*<br>*G. butleri* | Zygomycetes | Mucorales | Cunninghamellaceae | 58–216 | 89–92 | [52,60–65] |
| *Benjaminiella*<br>*B. poitrasii* | Zygomycetes | Mucorales | Mucoraceae | 51–78 | 94–95 | [66] |
| *Mucor*<br>*M. racemosus*<br>*M. rouxii*<br>*M. rouxianus*<br>*M. indicus* | Zygomycetes | Mucorales | Mucoraceae | 12–117<br>33–204<br>181<br>94–235 | 70–84<br>80–90<br>80–90<br>72–89 | [27,38,40,67–73] |
| *Rhizomucor*<br>*R. pusillus*<br>*R. miehei* | Zygomycetes | Mucorales | Mucoraceae | 80<br>14–137 | n.r.<br>81 | [71,74] |
| *Rhizopus*<br>*R. oryzae*<br>*R. oligosporus*<br>*R. arrhiizus* | Zygomycetes | Mucorales | Mucoraceae | 44–138<br>32<br>21–58 | 85–89<br>n.r.<br>82 | [51,52,55,68,75–81] |
| *Syncephalastrum*<br>*S. racemosum* | Zygomycetes | Mucorales | Syncephalastraceae | 74–152 | 72–77 | [27,82] |
| *Aspergillus*<br>*A. niger*<br>*A. terreus* | Eurotiomycetes | Eurotiales | Trichocomaceae | 70–209<br>69–141 | 81–90<br>85–88 | [28,38,68,83–86] |
| *Penicillium*<br>*P. chrysogenum*<br>*P. waksmanii* | Eurotiomycetes | Eurotiales | Trichocomaceae | 29–57<br>297 | 84–86<br>65 | [87–90] |

In fungi, both vegetative and sporulating cells are capable of chitin synthesis, and its secretion occurs in a polarized mode. As a matter of fact, chitin accumulation occurs at growth sites such as hyphal tips and cross-walls in filamentous fungi and emerging buds in yeasts. Chitin and chitosan accumulation mostly occurs in the cell wall's layers adjacent to the plasmalemma, where these glycans play a fundamental role in maintaining the cell wall's shape and integrity; moreover, they provide protection against foreign materials (e.g., cell inhibitors) and environmental stressors to which fungi might be exposed [91–94]. Owing to its positive charge, chitosan is capable of retaining anionic storage materials, such as polyphosphates, which are highly abundant in the Zygomycetes' cell wall [72]. Chitosan also exerts a role in some pathogenic fungi such as *Colletotrichum graminicola* [95] and *Magnaporthe oryzae* [96]. During infection by these species, the chitin deacetylase-catalyzed conversion of chitin into chitosan seems to preserve the appressorium from the hydrolysis by plant chitinases [95].

### 3.2. Fungal Producers of Chitin and Chitosan

In cell walls of fungal species belonging to the classes Deuteromycetes, Ascomycetes, and Basidiomycetes, chitin is regarded as the second most abundant component [97].

As opposed to chitin, chitosan is a less widespread cell wall constituent. However, Zygomycetes have significantly higher amounts of chitosan in their cell walls than other fungal classes. Within Zygomycetes, the most productive genera belong to the order of Mucorales and to two families, namely Cunninghamellaceae and Mucoraceae. The former family includes a variety of highly producing genera, such as *Absidia* [31,48,50], *Cunninghamella* [53,54,56,58] and *Gongronella* [60–62,64] while the latter includes the genera *Mucor* [27,40,67–73], *Rhizomucor* [71,74], and *Rhizopus* [68,76,77,80,81,98].

Although chitosan production from fungal sources can be regarded as a greener alternative to the shellfish-based process, it is not devoid of risks, depending on the selected fungal species. To exemplify, *Colletotrichum lindemuthianum* has been used as a valuable source of chitin deacetylase [99], although this fungus is the causative agent of 'anthracnose', a plant disease that can affect a variety of crops grown in both temperate and tropical climates. Within the class of Zygomycetes, some species belonging to *Absidia* and *Rhizopus* genera can be pathogens either to animals or humans. Some species of *Absidia* are causative agents of mucormycosis in humans with low immune systems [100] and zygomycosis, causing spontaneous abortion in cows. In addition, *Rhizopus oryzae* can act as an opportunistic human pathogen causing pulmonary mucormycosis [101]. As a consequence, the handling of these potentially pathogenic strains requires the adoption of specific measures to prevent their accidental dispersal. The hope is that the research and development of the chitosan fungal production process can be oriented in the future only and exclusively on strains that meet the generally regarded as safe (GRAS) requirement.

### 3.3. Production Processes of Fungal Chitin and Chitosan

The chitin and chitosan contents are species-specific and are largely affected by the growth medium and fermentation system. Although a variety of chitosan production processes have been conducted in solid-state fermentation (SSF) (Table 3), the majority of studies have relied primarily on liquid submerged fermentation (LsF) (Tables 4 and 5).

The marked preponderance of studies conducted in liquid fermentation is likely due to several advantages of this technique compared to SSF. These advantages include facile control of process parameters, especially at the reactor level, higher amenability to scale transfer, and straightforward biomass recovery from the growth medium [13]. In SSF, the fungal colonization occurs either on inert and homogeneous solid substrates or on natural ones moistened in such a way as to ensure the absence of free water [102]. On the one hand, using homogeneous solid substrates is advantageous as it allows better control of medium composition and improved oxygen and nutrients transfer; moreover, it facilitates the recovery of fungal biomass. However, no reports dealing with the chitin/chitosan production on inert substrates are available to the best of our knowledge. On the other hand, the natural use of solid substrates, due to their intrinsic heterogeneity, implies mass-transfer limitations, which do not enable accurate process control and enhance the difficulty of recovering fungal biomass [34].

Contrary to other reviews that have used a genus-specific approach for organizing literature data [3,103], a discussion based on the fermentation technique appeared to be more valuable and informative here (the present review). Considering the relative scarcity of studies on the solid-state production of chitosan and, in general, their poorly equipment-oriented character, only a single section of this review has been dedicated to this topic. Conversely, this review offers an articulated discussion of LSF production studies. These were divided based on the nature of the production medium into two sections, the first of which focused on studies conducted on synthetic media and the second on waste- or effluent-derived media. Finally, the present review devotes a section to process parameters that directly influence fungal chitosan production and properties. The reader is referred

to the conclusions section for any comparative considerations between the production approaches in liquid and solid-state culture.

### 3.3.1. Solid-State Production of Fungal Chitosan

As mentioned in Section 3.3, solid-state chitosan production studies have relied solely on naturally occurring substrates. This approach has offered the opportunity to explore economically sustainable production solutions and, at the same time, to exploit processing residues that would otherwise have little economic value. A variety of plant residues have been used for this purpose, including wheat and rice straw [104,105], soybean residues [76,106,107], hardwood sawdust [108], sweet potato pieces [60], potato chip processing waste [79], and cottonseed hulls [34].

Table 3, summarizing SSF chitosan production studies, shows that this technique enabled the achievement of high product concentration, referred to dry weight of solid substrate, which widely ranged from 1.6 to 17 g kg$^{-1}$ depending on strain and substrate combinations.

The average volumetric productivities ($r_P$) were also interesting in some cases, such as in cultures of *A. niger* and *M. rouxii* grown on soy-derived solid matrices (59 and 119 mg kg$^{-1}$ h$^{-1}$, respectively) [107,109]; these productivity values were attributable to a high product concentration rather than to rapid growth since the product peak was obtained in 12-day-old cultures in both cases.

**Table 3.** Chitosan production yields (CPY) referred to unit mass of the solid substrate, average volumetric productivity ($r_P$), degree of deacetylation, and viscosity of chitosan obtained in solid-state cultures of several fungal strains.

| Fungal Strain | Solid Substrate | Cultivation Mode | CPY (g/kg) | $r_P$ (mg/kg*h) | DD (%) | Viscosity (cP) | References |
|---|---|---|---|---|---|---|---|
| *Absidia coerulea* CTCC AF 93105 | Non-supplemented cotton seed hulls | Conical flask | 1.62 | 9.64 | 85 | n.r. | [34] |
| *Absidia coerulea* CTCC AF 93105 | Potato pieces added with sucrose and urea | Conical flask | 6.12 | 36.4 | 85 | n.r. | [34] |
| *Aspergillus niger* n.s. | Rice straw | Plastic bag with sterile filters | 5.26 | 24.4 | 84.2 | 59 | [105] |
| *Aspergillus niger* TISTR3245 | Mung bean residues | Conical flask | 1.39 | 19.3 | n.r. | n.r. | [76] |
| *Aspergillus niger* BBRC 20004 | Soybean residues | Conical flask | 17.03 | 59.1 | n.r. | n.r. | [109] |
| *Gongronella butleri* USDB0201 | Sweet potato pieces supplemented with urea | Tray reactor | 3.7 | 22.0 | 92–96 | n.r. | [61] |
| *Gongronella butleri* USDB0201 | Sweet potato pieces supplemented with urea | Tray reactor | 4.31 | 25.7 | n.r. | n.r. | [62] |
| *Lentinus edodes* SC-495 | Wheat straw | Plastic bag with sterile filters | 6.18 | 21.5 | 87.5 | n.r. | [104] |
| *Mucor rouxii* ATCC 24905 | Soybean meal | Autoclavable plastic bag with sterile filters | 32.4 | 119.4 | 55–60 | n.r. | [107] |
| *Penicillium citrinum* n.s. | Rice straw | Plastic bag with sterile filters | 5.12 | 17.8 | 78.5 | 4.6 | [105] |
| *Penicillium expansum* | Corn straw | Conical flask | 4.31 | n.r. | 80.2 | 4.8 | [110] |
| *Rhizopus oryzae* n.s. | Rice straw | Plastic bag with sterile filters | 5.63 | 19.6 | 90.2 | 6.8 | [105] |
| *Rhizopus oryzae* TISTR3189 | Potato peel | Conical flask | 6.6 | 55.0 | 87.5–90 | 3.1–6.1 | [79] |
| *Rhizopus oryzae* (local isolate) | Corn straw | Conical flask | 8.57 | 29.8 | 91.5 | 7.2 | [110] |
| *Rhizopus oryzae* TISTR3189 | Soybean residue | Conical flask | 4.3 | 29.9 | n.r. | n.r. | [76] |

Different evaluation, on the other hand, can be performed for potato- or sweet potato-derived substrates, which provided productivity values higher than those obtained on cereal residues owing to the comparatively lower time requirements to attain the product peak (Table 3). For example, the time required to achieve the product peak in *R. oryzae* TISTR3189 and *A. coerulea* CTCC AF 93105 cultures grown on solid potato-based matrices was 5 and 7 days [34,79]. Conversely, *L. edodes* SC-495, *R. oryzae*, and *P. citrinum* ATCC 24095 cultures grown on wheat straw, corn straw, and rice straw, respectively, reached the product peak 12 days after the inoculation [104,105,110].

Noteworthy, *Absidia coerulea* AF93105 solid-state cultures on potato waste provided a direct method of producing low molecular weight chitosan, which, due to its compatibility with agricultural and biomedical applications, is generally obtained by thermochemical or enzymatic depolymerization starting from high molecular weight chitosans [34]; in particular, the polymer obtained with a yield of 6.1 g kg$^{-1}$ showed an average molecular weight of 6.4 kDa associated with a very low degree of polydispersity.

Twelve-day-old *R. oryzae* solid-state cultures on nutrient-supplemented rice straw yielded 5.63 g chitosan kg$^{-1}$ substrate; the chitosan thus obtained exerted a higher antibacterial activity toward a variety of pathogenic bacteria, such as *Bacillus cereus*, *Pseudomonas aeruginosa*, *Salmonella* sp., and *Escherichia coli* as compared to chitosan from crab shells [105].

However, all these studies were performed either on stationary flasks or using autoclavable plastic bags. Although a variety of solid-state reactors are available, unique exceptions are the studies of one research group [60–62,111] that used a perforated tray reactor to perform chitosan production (3.7–4.3 g kg$^{-1}$) with excellent DD (92–96%) from *Gongronella butleri* USDB0201 cultures grown on potato peel wastes. An additional exception is the study of Dhillon et al. [3] who used a rotary tumbling drum reactor to investigate the coproduction of citric acid and chitosan.

### 3.3.2. Fungal Chitosan Production in Liquid Submerged Bioprocesses

Several screening studies aimed at identifying valuable chitosan-producing strains relied on synthetic media, such as the yeast extract-malt extract medium (YM) [45], potato dextrose broth (PDB) [68], yeast extract-peptone-dextrose (YPD) medium [27,46], glucose-peptone [88] and glucose-peptone-yeast extract (GPY) [75,112]. Moreover, a variety of strain-oriented studies, summarized in Table 4, used these production media in original or slightly modified formulation. Particularly in the last decade, the high costs of these media have shifted the attention of researchers toward cheaper solutions. Moreover, the temporal distribution of studies conducted on synthetic media shows that a non-negligible part of them is far from recent. This review, however, considered it appropriate to reserve them a section. The reasons underlying this choice are many and include a high relevance from the production point of view, the provision of information relating to the choice of strains, physiology of production, and reactor configuration.

#### Chitosan Production on Chemically Defined Liquid Media

Table 4 shows comparatively volumetric biomass and chitosan productions (X and CVP, respectively) and $r_P$ values of chitosan in liquid cultures of several fungal strains grown on synthetic media either in the shaken flask or in the reactor.

**Table 4.** Volumetric productions of biomass (X) and chitosan (CVP) and chitosan average volumetric productivity ($r_P$) in liquid cultures of several fungal strains grown on synthetic media either in shaken flask or in reactor.

| Fungal Strain | Culture Condition and Growth Medium | X (g L$^{-1}$) | CVP (g L$^{-1}$) | $r_P$ (mg L$^{-1}$ h$^{-1}$) | References |
|---|---|---|---|---|---|
| *Absidia butleri* NCIM977 | Shaken cultures on GYT medium | 6.78 | 0.57 | 7.9 | [51] |
| *Absidia coerulea* ATCC14076 | Aerated shaken cultures on YM broth | 6.2 | 1.86 | 26.0 | [47] |
| *Absidia coerulea* ATCC14076 | Batch cultures in 2.5 L STR at 250 rpm and 2 vvm with adaptive pH control at 4.5 on GY medium supplemented with $(NH_4)_2SO_4$ | 20 | 2.33 | 63.8 | [49] |

<div align="center">Table 4. *Cont.*</div>

| Fungal Strain | Culture Condition and Growth Medium | X (g L$^{-1}$) | CVP (g L$^{-1}$) | $r_P$ (mg L$^{-1}$ h$^{-1}$) | References |
|---|---|---|---|---|---|
| *Absidia coerulea* ATCC14076 | Batch cultures in 20 L STR at 200 rpm and 1 vvm on PGY medium | 13.9 | 0.55 | 11.5 | [113] |
| *Absidia coerulea* ATCC14076 | Continuous cultures in 2.5 L STR with pH control at 4.5 on GY medium supplemented with (NH$_4$)$_2$SO$_4$ at a dilution rate of 0.05 h$^{-1}$ | 7.0 | 1.04 | 50.0 | [49] |
| *Absidia coerulea* ATCC14076 | Continuous cultures in BioFlo C30 chemostat on GYP medium at a dilution rate of 0.025 h$^{-1}$ | 2.3 | 1.37 | 41.0 | [46] |
| *Absidia coerulea* CTCC AF 93105 | Shaken cultures on a glucose-based medium added with 0.5 g L$^{-1}$ as (NH$_4$)$_2$SO$_4$ | 11.4 | 2.86 | 19.9 | [50] |
| *Absidia coerulea* CCRC 30897 | Batch cultures in airlift with double-net draft tube on GYP medium | 30.8 | 3.16 | 65.8 | [114] |
| *Absidia coerulea* CCRC 30897 | Batch cultures in bubble column reactor on GYP medium | 11.3 | 1.36 | 28.3 | [114] |
| *Absidia glauca* (+) | Shaken cultures on GYP medium | 8.8 | 0.65 | 13.5 | [112] |
| *Absidia orchidis* NCAIM F 00642 | Batch cultures in 5 L STR on GYP medium supplemented with ferrous ions | 45.3 | 1.79 | 37.3 | [31] |
| *Absidia orchidis* NCAIM F 00642 | Batch cultures in 5 L STR on GYP medium supplemented with Mn$^{2+}$ ions | 15.2 | 1.05 | 21.9 | [31] |
| *Absidia repens* CBS 102-32 | Batch cultures in 10 L STR at 350 rpm on a medium made of glucose, yeast extract, and (NH$_4$)$_2$SO$_4$ | 12.9 | 2.8 | 58.3 | [115] |
| *Aspergillus niger* MTCC 872 | Shaken cultures on a medium made of potato dextrose broth (24 g L$^{-1}$), glucose (80 g L$^{-1}$), L-Asparagine (6 g L$^{-1}$) | 15.9 | 3.35 | 46.5 | [84] |
| *Aspergillus niger* BBRC 20004 | Shaken cultures on Sabouro dextrose broth added with 2% glucose | 5.17 | 0.84 | 17.5 | [83] |
| *Aspergillus nidulans* NS | Shaken cultures on peptone-glucose-yeast extract (PGY) salt broth | 5.15 | 0.20 | 4.19 | [112] |
| *Benjaminiella poitrasii* CSIR isolate | Batch cultures in 2 L STR on medium containing (g L$^{-1}$): yeast extract, 6.0; peptone, 10.0; soluble starch, 10.0 | 10.0 | 0.51 | 10.6 | [66] |
| *Cunninghamella bertholletiae* IFM 46.114 | Shaken cultures on yeast extract-peptone-dextrose medium | 7.1 | 0.39 | 5.5 | [56] |
| *Cunninghamella echinulata* | Shaken cultures on glucose-peptone-yeast extract medium added with (NH$_4$)$_2$SO$_4$ | 5.6 | 0.40 | 3.3 | [52] |

**Table 4.** *Cont.*

| Fungal Strain | Culture Condition and Growth Medium | X (g L$^{-1}$) | CVP (g L$^{-1}$) | $r_P$ (mg L$^{-1}$ h$^{-1}$) | References |
|---|---|---|---|---|---|
| *Cunninghamella elegans* IFM 46109 | Shaken culture on a medium mainly made of glucose, asparagine, and MgSO4 (60, 3.0, and 0.25 g L$^{-1}$, respectively) | 11.0 | 0.86 | 8.9 | [53] |
| *Cunninghamella elegans* UCP 542 | Shaken cultures on Sabouraud-sucrose medium | 12.0 | 0.42 | 8.7 | [57] |
| *Gongronella butleri* USDB 0201 | Shaken cultures on glucose-peptone-yeast extract medium added with (NH$_4$)$_2$SO$_4$ | 8.2 | 0.47 | 3.9 | [52] |
| *Mucor racemosus* (soil isolate) | Shaken cultures on Sabouraud dextrose broth | 3.8 | 0.45 | 2.6 | [71] |
| *Mucor rouxii* ATCC 24905 | Shaken cultures on glucose-peptone-yeast extract medium | 3.8 | 0.28 | 5.8 | [67] |
| *Mucor rouxii* ATCC 24905 | Shaken cultures on peptone-yeast extract-glucose (PYG) salt broth | 5.6 | 0.21 | 4.4 | [112] |
| *Mucor rouxii* DSM 1191 | Batch cultures in 30 L STR on glucose-peptone-yeast extract medium | 8.6 | 0.30 | 21.2 | [116] |
| *Rhizomucor miehei* ATCC 26282 | Shaken cultures on Sabouraud dextrose broth | 4.1 | 0.56 | 3.4 | [71] |
| *Rhizopus oryzae* USDB 0602 | Shaken cultures on glucose-peptone-yeast extract medium added with (NH$_4$)$_2$SO$_4$ | 5.7 | 0.28 | 2.3 | [52] |
| *Syncephalastrum racemosum* UCP148 | Shaken cultures on yeast extract-peptone-dextrose medium | 8.0 | 1.26 | 26.1 | [27] |

By sorting the data shown in Table 4 based on the volumetric production, it is evident, with a few exceptions, that the best performing strains belong to the *Absidia* genus. This outcome does not change using the $r_P$ of chitosan as the sorting criterion, and it is no coincidence that 8 of the 10 highest values are related to studies conducted in reactors. Furthermore, it is in agreement with the investigation of Shimahara et al. [117], who had already concluded several years earlier at the end of a screening conducted on 125 strains of Zygomycetes, that those belonging to the genus *Absidia* were by far the most productive.

Wu et al. [114] achieved the best $r_P$ (65.8 mg L$^{-1}$ h$^{-1}$) ever reported with *A. coerulea* CCRC 30897 batch cultures grown in an airlift reactor modified with a double-net draft tube. This modified reactor enabled an excellent oxygen transfer to the liquid medium resulting in the achievement of a CVP value more than two times higher than that observed in a conventional bubble column reactor (3.16 vs. 1.36 g L$^{-1}$). The same study also compared the performance of the modified airlift with that of a mechanically agitated reactor and found that CVP values were more than two-fold and 55% higher than those achieved in an STR with an impeller speed of 600 and 300 rpm, respectively.

Kim et al. [49] also obtained very relevant results with another *A. coerulea* strain, viz. 14076, grown in a 2.5 L STR operated either in batch or continuous mode and using a glucose-YE medium supplemented with (NH$_4$)$_2$SO$_4$. In this study, the strategy of pH control at 4.5 led to a higher maximal growth rate and smaller pellets than cultures where pH was left to fluctuate freely, leading to 1.8- and 3.5-fold improvement in CVP and $r_P$, respectively. The same study claimed that when the STR was operated in continuous mode at a dilution rate of 0.05 h$^{-1}$, the $r_P$ of the process was 52 mg L$^{-1}$ h$^{-1}$, a value higher than

that obtained in a chemostat with the same strain by Rane and Hoover [46]. Noteworthy, batch cultures of the same strain grown in a 20 L STR in a similar medium [113] exhibited lower performance than those of Kim et al. [49], presumably owing to either the larger process scale or omitted control of medium's pH.

Another species belonging to the genus *Absidia* that proved to be a suitable producer of chitosan was *A. repens* CBS 102.32; this strain, grown in a 10 L STR on a medium consisting of glucose and YE and added with $(NH_4)_2SO_4$, provided CVP and $r_P$ values equal to 2.8 g $L^{-1}$ and 58.3 mg $L^{-1}$ $h^{-1}$, respectively [115]. Although the CVP value was moderate (0.3 g $L^{-1}$), the study by Gözke et al. [116], conducted in a 30 L STR reactor with *M. rouxii* DSM 1191 cultures on GYP medium, is noteworthy due to the scale of the reactor and the short process duration reaching maximum productivity after 14–16 h from the inoculation. The use of inorganic supplements, such as manganese and ferrous ions able to affect the chitin synthase and chitin deacetylase activities was tested by Jaworska and Konieczna [31]. This study was conducted in a 4.5 L STR with *A. orchidis* NCAIM F 00642 grown on iron-supplemented yeast extract-peptone glycerol medium led to CVP and $r_P$ values equal to 1.79 g $L^{-1}$ and 58.3 mg $L^{-1}$ $h^{-1}$, which were several folds higher than those of non-supplemented cultures.

Chitosan Production on Waste- and Effluent-Based Liquid Media

As discussed in the previous section, the liquid media for fungal chitosan production often include organic components, such as YE, D-glucose, and peptone, which are costly growth substrates. For this reason, several studies investigated the exploitation of cheap carbon and nitrogen sources derived from wastes [64,81,86,90] to mitigate the production costs and compete commercially with crustacean's shell-based processes [43].

With regard to their origin, these wastes derived from crop residues [118], corn wet-milling operations [119] or were byproducts of the dairy industry [77,78], sugar manufacturing [56,70,81,120], fruit juice industry [64,65] or distilleries [121,122]. In several cases, the liquid medium was derived from a solid substrate either by acid hydrolysis, such as for corn straw [118], or via its aqueous extraction, such as in the case of apple pomace [64,86] and date syrups [28]. Other studies, instead, relied on liquid byproducts, such as sugarcane or sugar beet molasses [56,81,82] and deproteinized whey [77]. Suitable liquid production media have also been derived from a variety of process effluents, such as cassava wastewater [54], paper mill effluent [90], thin stillage [122], and xylose-rich wastewater from a bioethanol plant [74] (Table 5).

In several cases, the approach adopted involved only a partial replacement of the expensive organic components of the medium, intended to act as sources of carbon or nitrogen, with others derived from residues of agro-industrial origin. For example, Jiang et al. [50] obtained excellent CVP (4.11 g $L^{-1}$) and $r_P$ (28.54 mg $L^{-1}$ $h^{-1}$) values in *A. coerulea* CTCC AF 93105 cultures on a glucose-based medium by replacing commercial sources of inorganic nitrogen with soybean pomace.

**Table 5.** Volumetric productions of biomass (X) and chitosan (CVP) and chitosan average volumetric productivity ($r_P$) in liquid cultures of several fungal strains grown on liquid byproducts or process effluents either in shaken flask or in reactor.

| Fungal Strain | Culture Condition and Growth Medium | X (g $L^{-1}$) | CVP (g $L^{-1}$) | $r_P$ (mg $L^{-1}$ $h^{-1}$) | References |
|---|---|---|---|---|---|
| *Absidia coerulea* CTCC AF 93105 | Shaken cultures on a glucose-based medium (20 g $L^{-1}$) added with 0.5 g $L^{-1}$ nitrogen as soybean pomace | 15.4 | 4.11 | 28.5 | [50] |
| *Aspergillus awamori* MTCC6995 | Shaken cultures on thin stillage from rice-based distillery | 5.2 | 0.39 | 4.0 | [122] |

**Table 5.** *Cont.*

| Fungal Strain | Culture Condition and Growth Medium | X (g L$^{-1}$) | CVP (g L$^{-1}$) | $r_P$ (mg L$^{-1}$ h$^{-1}$) | References |
|---|---|---|---|---|---|
| *Aspergillus braziliensis* ATCC16404 | Shaken cultures on syrup from date waste | 13.3 | 2.78 | 19.3 | [28] |
| *Aspergillus niger* NRRL567 | Batch cultures in 7.5 L STR (impeller speed at 200 rpm; adaptive flow rate to ensure 20% dissolved oxygen saturation) on apple pomace sludge | 12.6 | 0.64 | 4.9 | [3] |
| *Cunninghamella bertholletiae* IFM 46.114 | Shaken cultures on sugarcane juice (10.5 g L$^{-1}$ sucrose) added with YE (3 g L$^{-1}$) | 4.2 | 0.53 | 11.1 | [56] |
| *Cunninghamella elegans* UCP 0542 | Shaken cultures on a medium made of cassava wastewater (CWW, 10%) and corn steep liquor (CSL, 4%) | 5.7 | 0.33 | 4.6 | [54] |
| *Gongronella butleri* CCT4274 | Shaken cultures on aqueous extract of apple pomace supplemented with NaNO3 (2.5 g L$^{-1}$) | 5.5 | 1.19 | 16.4 | [64] |
| *Gongronella butleri* IFO8081 | Shaken cultures on sweet potato shochu distillery wastewater | 6.2 | 0.73 | 6.1 | [121] |
| *Gongronella butleri* CCT 4274 | Batch cultures in 6.5 L airlift reactor with external loop circulation (aeration rate, 0.6 vvm) on apple pomace extract added with 5 g L$^{-1}$ (NH$_4$)$_2$SO$_4$ | 6.7 | 0.93 | 62.2 | [65] |
| *Lichtheimia hyalospora* UCP1266 | Shaken cultures on a medium made of CWW (4%) and CSL (6%) | 11.9 | 0.75 | 6.3 | [119] |
| *Mucor rouxii* MTCC 386 | Shaken cultures on molasses salt medium added with indole-3-acetic acid (1.0 mg L$^{-1}$) | 9.1 | 0.95 | 29.7 | [70] |
| *Mucor subtilissimus* UCP 1262 | Shaken cultures on a medium made of CWW (4%) and CSL (6%) | 4.8 | 0.16 | 1.3 | [119] |
| *Penicillium citrinum* (local isolate) | Batch cultures in 3 L stirred tank reactor (200 rpm and 2.0 vvm) on paper mill effluent added with 50 mg L$^{-1}$ acetic acid | n.s. | 0.14 | 2.9 | [90] |
| *Rhizopus arrhizus* UCP 0402 | Shaken cultures on a medium made of CSL (4%) and honey (13%) | 11.7 | 0.34 | 3.6 | [80] |
| *Rhizopus oryzae* MTCC262 | Shaken cultures on deproteinized whey added with gibberellic acid (0.1 mg L$^{-1}$) | 8.3 | 1.13 | 15.7 | [77] |
| *Rhizopus oryzae* 00.4367 | Batch cultures in 7 L STR (340 rpm, 2.1 vvm) on untreated sugarbeet molasses (45.4 g L$^{-1}$ total sugars) | 10.7 | 1.06 | 14.7 | [120] |

**Table 5.** *Cont.*

| Fungal Strain | Culture Condition and Growth Medium | X (g L$^{-1}$) | CVP (g L$^{-1}$) | $r_P$ (mg L$^{-1}$ h$^{-1}$) | References |
|---|---|---|---|---|---|
| *Rhizopus oryzae* AS 3.819 | Batch cultures in 3 L stirred tank reactor (200 rpm, 1 vvm) on corn stover hydrolysate supplemented with urea (4 g L$^{-1}$) | 11.0 | 0.99 | 13.8 | [123] |
| *Rhizopus oryzae* PAS 17 | Shaken cultures on medium made of molasses (7%, *v/v*) and supplemented with MgSO$_4$ | 10.7 | 1.50 | 7.8 | [81] |
| *Rhizopus oryzae* ME-F12 | Shaken cultures on corn straw hydrolysate | 5.2 | 0.58 | n.s. | [118] |
| *Rhizopus oryzae* MTCC262 | Shaken culture on deproteinized whey supplemented with (NH$_4$)$_2$HPO$_4$ (8 g L$^{-1}$) and YE (2 g L$^{-1}$) | 6.2 | 0.62 | n.s. | [78] |
| *Syncephalastrum racemosum* UCP148 | Shaken cultures on sugarcane juice (10.5 g L$^{-1}$ sucrose) added with YE (3 g L$^{-1}$) | 8.1 | 0.60 | 5.0 | [82] |
| *Syncephalastrum racemosum* UCP148 | Batch cultures in 5 L STR on sugarcane juice (10.5 g L$^{-1}$ sucrose) added with YE (3 g L$^{-1}$) | 8.0 | 0.96 | 32.0 | [82] |

*S. racemosum* UCP148 cultures grown in a 5 L STR containing sugarcane juice supplemented with 0.3% YE yielded CVP and $r_P$ values as high as 0.93 g L$^{-1}$ and 32 mg L$^{-1}$ h$^{-1}$, respectively [82].

In other studies, instead, the approach was to integrally eliminate the expensive organic components of the medium with the liquid residue used as it was or, possibly, supplemented with inorganic nitrogen sources. For instance, *G. butleri* CCT 4274 cultures grown in a 6.5 L airlift reactor on an aqueous extract of apple pomace (AEAP) supplemented with an inexpensive source of nitrogen, such as ammonium sulfate (5 g L$^{-1}$) provided a CVP value of 0.93 g L$^{-1}$ but, above all, one of the highest $r_P$ values of chitosan ever reported for reactor cultures (62 mg L$^{-1}$ h$^{-1}$) [65]. The use of the airlift enabling better gas exchanges associated with the use of a more readily available nitrogen source provided much better results than those reported by Streit et al. [64] with the same strain grown on sodium nitrate-supplemented AEAP. Göksungur [120] used untreated and non-supplemented sugar beet molasses as the liquid medium for chitosan production by *R. oryzae* 00.4367 in a 7 L STR; the use of a response surface methodology approach allowed the investigators to optimize statistically both agitation and aeration regimes and sugar concentration. In particular, under the optimization of these variables (impeller speed, 340 rpm; aeration rate, 2.1 vvm; sugar concentration, 45.4 g L$^{-1}$), the CVP and $r_P$ amounted to 1.06 g L$^{-1}$ and 14.5 mg L$^{-1}$ h$^{-1}$, respectively.

Whey, a byproduct of the dairy industry, the world production of which amounts to 121 million tons [124], was also used as the basis for the development of a chitosan production medium. In particular, *R. oryzae* MTCC262 cultures, grown on deproteinized whey (DW) and added with 0.3% YE and 0.1 mg L$^{-1}$ of gibberellic acid (GA3), provided CVP and $r_P$ values equal to 1.13 g L$^{-1}$ and 15.72 mg L$^{-1}$ h$^{-1}$ [77]. In a subsequent study, the same research group reported for the same strain on a DW-based medium, but differently formulated (Table 5), CVP and $r_P$ values equal to 0.62 g L$^{-1}$ and 8.6 mg L$^{-1}$ h$^{-1}$ [78].

With regard to the use of process effluents for the development of chitosan production media, several studies focused their attention on that produced by the cassava (*Manihot esculenta*) processing industry, termed cassava wastewater (CWW). Recent estimates es-

tablished that the processing of 1 ton of tubers generates around 60 m$^3$ of effluent [125]. Several studies shared the combination of CWW with corn steep liquor, a byproduct of corn wet-milling, frequently used as a low-cost nitrogen source [54,63,119]. Among these studies, the best process performance was obtained with *Lichtheimia hyalospora* UCP1266 yielding CVP and $r_P$ values amounting to 0.75 g L$^{-1}$ and 4.56 mg L$^{-1}$ h$^{-1}$ [119].

Grain-based distilleries generate thin stillage (TS) as the high-strength process effluent, the volume of which is ten-fold higher than that of the ethanol produced. Ray and Ghangrekar [122] used *Aspergillus awamori* MTCC6995 to reduce the organic load of a rice-based TS and to extract chitosan from the residual fungal biomass; at its endpoint (96 h), the process yielded a 60% reduction in the effluent's COD and CVP and $r_P$ of chitosan amounted to 0.39 g L$^{-1}$ and 4.02 mg L$^{-1}$ h$^{-1}$, respectively. Yokoi et al. [121] used sweet potato shochu wastewater as the chitosan production medium for *G. butleri* IFO 8081, which attained a CVP of 0.73 g L$^{-1}$ after 120 h from the inoculation.

### 3.4. Relevant Factors in Chitin and Chitosan Production from Fungi

Production levels of these glycans can be achieved via an increase in the biomass yield or an increase in their contents in the cell wall [31], and several process parameters were found to be relevant to this goal. These parameters affect not only the production of these glycans but also their physicochemical properties [31,69,77].

### 3.4.1. Fungal Morphology

Fungal morphology, especially during cultivation in reactors, frequently evolves in a way resulting in a decline in the growth rate [126]. The growth mode involving the formation of dispersed mycelium leads to a highly viscous medium with ensuing agitation and aeration problems [127]. Several strains, instead, have a marked propensity to form pellets, and, in that instance, an increased pellet diameter might result in dropped growth owing to diminished mass transfer to the pellet's innermost part [126]. Several studies highlighted the importance of the pellet size, showing that diameters in the 4.0–5.0 mm were most conducive to the maximization of D-glucosamine yield in *R. oligosporus* [128] or chitosan production by *Lichtheimia hyalospora* UCP 1266 [119]. Similar results were obtained with *Absidia repens* CBS 102.32 [115], grown at different stirring regimes (i.e., 350, 200, 200 for the early 24 h followed by 350 rpm) in a 10 L STR; growth at 350 rpm, leading to the formation of pellets with an average diameter of 0.5 mm was the condition enabling best chitosan production (2.8 g L$^{-1}$) with an $r_p$ of 58 mg L$^{-1}$ h$^{-1}$.

Macro-morphology can also largely affect chitosan production of several dimorphic species, such as *M. indicus* [129] and *M. subtilissimus* [119]. The growth mode in dimorphic fungi is affected by several factors, including initial spore concentration, sugar content in the growth medium, and oxygen availability; moreover, the shift to a yeast-like morphology can be promoted by the addition of compounds acting as cytochrome oxidase inhibitors. Noteworthy, de Souza et al. [119] reported that the increase in concentrations of cassava wastewater, containing cyanides, promoted a yeast-like growth mode and negatively affected the chitosan production in *M. subtilissimus* UCP 1262 cultures. The same study found that the best chitosan production conditions were those leading to the mycelial form in agreement with other studies conducted with *M. indicus* CCUG 22424 [129] and *M. rouxii* [130].

### 3.4.2. Harvesting Time

Although chitin and chitosan productions are obviously growth-associated processes, their maximum yields do not necessarily occur at the biomass peak [4]. With this regard, in fact, the amount of extractable chitosan was highest at the late exponential growth phase in *Asidia coerulea* [50], *Rhizopus oryzae* 00.4367 [120], *R. oryzae* USDB 0602 [52], and *Cunninghamella bertholletiae* [56] to decline significantly thereafter. Tan et al. [52] suggested that free chitosan molecules were largely abundant during the exponential phase, owing to active growth; during the stationary growth phase, instead, a higher proportion of

chitosan bound to other cell's wall constituents, thus rendering extraction less effective. This effect was also evident in *R. oryzae* solid-state cultures on rice straw, where the ratio between chitosan and alkali-insoluble material (AIM) dropped from 61%, in concomitance with biomass peak, to 42% when culture had entered the stationary phase [105]. Albeit not yet proven in vivo, an alternative explanation might be a time-dependent increase in chitin crystallinity due to the failure of the majority of CDAs to perform the in vitro deacetylation of crystalline chitin [131]. A notable exception to this time-dependent trend in chitosan production was the study of Davoust and Persson [115], who observed chitosan accumulation in the late stationary phase of *Absidia repens* cultures; they hypothesized that since the synthesis of chitin and chitosan occurs at the hyphal apex during the elongation process, the observed increase was due to continued apical growth at the expense of products derived from partial cell lysis.

### 3.4.3. Medium's Ph

By using a modified GPY medium adjusted to several pH values (range: 3.0–6.5), Rane and Hoover [46] found that the chitosan-producing ability and DD of chitosan were largely unaffected in *M. rouxii* DSM1191; the same study, however, found a significant strain-dependent effect of the pH on the same descriptors in *A. coerulea* ATCC 14076 and *A. coerulea* NRRL 1315. Kim et al. [49] observed that pH affected fungal morphology in *A. coerulea* ATCC 14076 cultures, and adaptive control at pH 4.5 enabled the formation of smaller pellets than those without pH control in batch cultures conducted in an STR. This pH control strategy enabled higher CVP and $r_\text{P}$ as compared to cultures without pH control (2.3 vs. 1.3 g L$^{-1}$ and 63.9 vs. 36.1 mg L$^{-1}$ h$^{-1}$, respectively). Solid-state *G. butleri* cultures conducted on sweet potato pieces at various initial pH (i.e., 3.77, 4.92, 5.46, and 5.52) provided better chitosan yields at pH 5.46 and 5.52, and the chitosans obtained had higher average molecular weights than those at more acidic pHs [62].

### 3.4.4. Nitrogen Source and Concentration

Both the nitrogen source and its amounts are among the most relevant process parameters since chitosan is a nitrogen-containing glycan. In general, fungi are able to directly exploit ammonium ions, while other inorganic nitrogen sources have to be reduced to the redox level of ammonium [132]. Nwe and Stevens [62] showed that urea was a valuable N source for chitosan production by *Gongronella butleri* solid-state cultures. By increasing the urea levels from 5 to 14 g per kg of solid substrate, in addition to obtaining a significant increase in the yield of chitosan (from 0.082 to 0.114 g g$^{-1}$ mycelium), there was a disproportionate increase in weight average molecular weight (Mw) as compared to number average molecular weight (M$_\text{n}$), thus increasing polydispersity (Mw/M$_\text{n}$) [62]. A dose-dependent effect of the N source on the molecular weight of *Mucor rouxii* chitosan was also observed by Arcidiacono and Kaplan [33] by doubling either peptone or yeast extract concentration in a YPG medium. Several studies claimed that the impact of yeast extract, a costly organic nitrogen source derived from cells autolysis, was beneficial to chitosan production [33,56,82]. To improve the economic feasibility of the chitosan production process, Abasian et al. [73] replaced YE with an autolysis-derived *M. rouxii* extract in a glucose-based medium; this approach improved the yields of the AIM and led to increased GlcN and decreased GlcNAc concentrations in AIM in *M. rouxii* CCUG 22424 cultures.

### 3.4.5. Plant Growth Hormones

Some studies showed that the addition of phytohormones to *R. oryzae* and *Mucor rouxii* liquid cultures grown on deproteinized whey [77] and molasses salt medium [70], respectively, stimulated both fungal biomass and chitosan productions although in a dose-dependent manner. Among the tested phytohormones (i.e., indole-3-acetic acid, IAA; indolebutyric acid, IBA; gibberellic acid, GA3 and kinetin, KIN), GA3 was the most effective. In fact, the addition of GA3 at 0.1 and 3.0 mg L$^{-1}$ concentration to *R. oryzae* [77] and *M. rouxii* [70] cultures resulted in a 50% and 69% increase in chitosan, respectively;

however, in both studies, higher GA3 doses had an inhibitory effect on chitosan production. Chatterjee et al. [77] also showed a 27% increase in the specific activity of chitin deacetylase in GA3-added cultures as compared to control ones, thus leading to an increased chitosan/chitin ratio (0.72 vs. 049). Another impact of GA3 addition was an increase in MW of chitosan derived from hormone-added cultures as compared to control ones [70,77].

### 3.4.6. Organic Stimulators

Another range of potential stimulators of chitosan biosynthesis emerged indirectly from a study reporting on the superiority of a corn straw hydrolysate for chitosan production by *R. oryzae* ME-F12 over glucose-, and xylose-based media [118]. This acid hydrolysate, in addition to containing xylose, as the main component and other pentoses, also contained furfural, acetic acid, and formic acid. The last three compounds added to a xylose medium exerted a dose-dependent stimulatory effect on both chitosan production and chitosan yield [118]; the authors speculated that the enhanced chitosan synthesis was a compensatory response of the fungus to increase the thickness and density of the cell wall to better cope with the presence of potentially inhibitory compounds.

### 3.4.7. Inorganic Supplements

Some studies have taken into consideration the effect deriving from the addition of some inorganic supplements, such as $Mg^{2+}$, $Co^{2+}$, $Mn^{2+}$ and $Fe^{2+}$ ions, capable of influencing the in vitro activity of chitin deacetylase and chitin synthetase [131,133]. For instance, the wide recognition of the stimulatory effect of $Mg^{2+}$ ions on both CS and CDA have led to the inclusion of $MgSO_4$ in several chitosan production media at a concentration ranging from 0.5 to 5.0 g $L^{-1}$ [31,46,49].

For some microelements such as cobalt, the effect was strongly concentration dependent or even led to growth failure due to the intrinsic toxicity of this metal. In fact, Rane and Hoover [46] reported that while the addition of 5 mg $L^{-1}$ to a modified GPY medium led to a 20% increase in chitosan production in *A. coerulea* ATCC 14076 cultures, a four-fold increase in its concentration led to inhibitory effects. In another study attempting the exploit the stimulatory effect of several ions, the addition of $Co^{2+}$ (2.3 and 4.5 g $L^{-1}$) to a YPG medium led to severe growth inhibition in *A. orchidis* NCAIM F 00642 cultures [31]. Conversely, the same study found that the addition of $Mn^{2+}$ ions led to an improved CVP as compared to control cultures (1.03 vs. 0.71 g $L^{-1}$), increased viscosimetric molecular weight (1156 vs. 751 kDa), and decreased DD (69.3% vs. 84.4%).

Phosphates are the primary constituents in the cell walls of Zygomycetes, and several studies indicate that their contents can vary from 8.3% to 23% [130,134]. Phosphates interact with chitosan and other structural polysaccharides to yield complexes that are not broken down by conventional acid treatment. As a consequence, a significant amount of chitosan can remain associated with both acid- and alkali-insoluble materials during extraction. Several studies showed that low concentrations of phosphates in the growth medium led to increased GlcN yields [39,135]. In Zygomycetes, the accumulation of anionic storage materials, such as polyphosphates, is thought to be performed by chitosan due to its polycationic nature [72]. Consequently, under P-limiting conditions, chitosan synthesis is probably boosted to enhance the uptake of phosphates from the growth medium. Moreover, in media containing inorganic nitrogen sources, such as ammonium sulfate (AS), the limitation of inorganic phosphorus (Pi) stimulates the overproduction of chitin/chitosan in some Zygomycetes [136]. Acidic stress caused by the low concentrations of Pi in the growth medium was suggested to be the primary reason explaining this phenomenon in AS-Pi media at low Pi concentrations [137]. These findings are in agreement with a chitosan optimization study conducted with *Mucor indicus* CCUG22424 cultures, where a phosphate-free medium led to the highest chitosan production [39].

## 4. Integrated Bioprocesses

In recent decades, significant advances have been made toward promoting a sustainable bio-based economy. A positive outcome of these endeavors primarily depends on the enactment of the concept of biorefinery, namely a facility able to integrate processes and equipment to yield a wide array of marketable products and energy [138]. Pursuing the concept of coproduction can increase the economic viability of the microbial processes currently in place [3,136]. Moreover, there is an undeniable urgency to promote integrative technology to exploit the vast amounts of mycelial waste from industrially relevant processes. In this framework, the mycelial wastes from various bioprocesses can be exploited as feasible chitosan sources in compliance with the biorefinery concept. These fungal strains mostly belong to the *Aspergillus* and *Mucor* genera, the species of which are used in several relevant processes. Unfortunately, microbial productions are sometimes governed by very different factors, and, therefore, the concept of coproduction can be difficult to pursue. This may mean sacrificing one of the two coproducts or, otherwise, using tools capable of safeguarding the production levels of both. Among these tools, we can mention the use of high-throughput culture techniques [136] and that of statistical optimization methods enabling the identification of variable combinations maximizing both coproducts, such as in response surface methodology [13,40].

In a very recent study, the combination of high-throughput culture technique with fast Fourier transform infrared spectroscopy allowed a fast determination of the impact of two relevant variables (i.e., two nitrogen typologies and inorganic phosphate) on the coproduction of single-cell oils, polyphosphates, and chitin/chitosan in nine different Zygomycetes strains [136].

One relevant bioprocess is the annual production of citric acid by *Aspergillus niger* strains, estimated to be around 1.7 million tons and generating 0.34 million tons of fungal waste [3]. The average chitin contents in *A. niger* strains are around 15–22% of the dry mycelium, and the polymer can be easily extracted and converted into chitosan [139]. Dhillon et al. [3] integrated citric acid (CA) production by *A. niger* NRRL567 cultures on apple processing wastes and subsequent extraction of chitosan from the waste stream; under LsF conditions, 132 h-old cultures grown in a 7.5 L STR on apple pomace sludge, provided volumetric CA and chitosan productions of 18.4 and 0.64 g L$^{-1}$, respectively. Solid-state cultures on apple pomace performed well, leading to CA and chitosan production of 182 and 64 g kg$^{-1}$ dry substrate, respectively, after 120 h from the inoculation [140]; the viscosity (1.02–1.18 mPa s$^{-1}$) and DD (78%–86%) of chitosan preparations from SSF and LSF cultures resembled those of chitosan from crab shells. A previous study conducted on the *A. niger* spent biomass from a production plant of citric acid showed that the use of an enzymatic extraction method yielded chitosan with higher MW and D-glucosamine content and a similar DD as compared to the conventional alkali-acid reflux method [139].

Liao et al. [98] investigated the coproduction of fumaric acid and chitin by *Rhizopus oryzae* ATCC 20344 using a N-nitrogen-rich liquid fraction of dairy manure and a DM-derived hydrolysate from sequential alkaline peroxide and enzymatic treatment; under optimal conditions, volumetric fumaric acid and chitin productions were 31 and 2.4 g L$^{-1}$, respectively.

The lipid-accumulation ability of several members of the Mucorales order makes them valuable candidates in second-generation biodiesel production [141]. This feature, combined with the greater proportion with which chitosan and chitin enter the constitution of the cell wall, opens up the possibility of profitably using their residual biomass. Zininga et al. [40] seized this opportunity by associating the production of biodiesel by *M. circinelloides* ZKT with the chitosan recovery from the spent biomass; the lipid and chitosan contents of the biomass grown on a glucose-YE-peptone medium were 21.4% and 11.2%, respectively.

Vinche et al. [142] developed an anaerobic chitosan-ethanol coproduction process by *Rhizopus oryzae* cultures grown on wheat hydrolysate containing variable glucose amounts (15–190 g L$^{-1}$). The initial sugar concentration markedly affected chitosan and ethanol

production. The yield of the latter ranged from 0.28 to 0.46 (g g$^{-1}$ sugar consumed), and the best value of this parameter was observed at 45 g L$^{-1}$ glucose. The AIM was a major fraction (17–25%) of the *R. oryzae* dry mycelium, and GlcN and GlcNAc constituted 61–65% of this fraction. In particular, the GlcN content reached its maximum (0.46 g g$^{-1}$ AIM) at a glucose concentration of 45 g L$^{-1}$.

*Penicillium chrysogenum* is extensively exploited for the industrial production of antibiotics, thus generating large amounts of mycelial wastes mostly disposed of by landfilling or incineration. On average, the production of 1 ton of penicillin generates 8–10 tons of waste biomass, the annual amount of which is estimated to amount to 1.2 million tons [143].

*P. chrysogenum* mycelial waste supplied by a pharmaceutical company [144] underwent an integrative extraction method aimed at recovering (1-3)-α-D-glucan, ergosterol, and chitosan; since the presence of alkali was required for the chitin deacetylation and for saponification during ergosterol isolation, the extraction scheme integrated these two steps for simplification.

## 5. Conclusions

There are several companies on the market that commercialize chitosan-based products of fungal origin, such as Kitozyme (http://www.kitozyme.com; accessed on 30 December 2021), MycoDev Group Inc. (https://mycodevgroup.com; accessed on 30 December 2021) and Chibio (https://www.chibiotech.com; accessed on 30 December 2021). This suggests an autonomous production capacity or, alternatively, the exploitation of fungal biomass derived from mycobiotechnological processes. Although the scientific literature currently available witnesses several efforts toward process upscaling to a diverse range of LSF reactors, studies reporting process scales on the order of cubic meters are absent. The solid-state production of chitosan has been regarded as a promising route since it enables high product concentration and is less equipment oriented than LsF. However, the use of SSF is negatively affected by mass-transfer limitation phenomena, heat dissipation problems above all, which become critical in large-scale processes. Although SSF seems to provide better performance than LSF in some cases [104,109], a non-negligible factor is the low bulk density of several solid substrates, which negatively affects the loadable mass inside the reactor. Therefore, comparisons between SSF and LSF, based on mass balances, should be made considering the masses of solid and volumes of liquid media, respectively, that can be used for the same working capacity of the reactor.

Despite these limitations, there are several issues that witness in favor of fungal chitosan. The advantages of fungal approaches to chitosan production are well documented, with relatively uniform physicochemical properties of the product made possible by accurate bioprocess control. As discussed earlier, a variety of studies shows that MW, polydispersity, and DD can be manipulated by deliberate variations in process conditions. Obtaining reproducible values of these parameters is fundamental to guaranteeing the acceptance of chitosan in critical sectors such as the medical and pharmaceutical sectors. The possibility of controlling MW and polydispersity of the polymer allows satisfying specific application requirements. This target is made difficult with the conventional extraction process relying on shellfish wastes due to the variability and non-uniformity of the raw material and the relative complexity of the extraction scheme. Switching from conventional to mushroom-based processes requires economic and environmental factors to be balanced and carefully evaluated. A likely scenario involves an increased environmental unacceptability of the conventional process associated with raised pollution abatement costs. An ever-increasing adoption of integrated bioprocesses that, in addition to the primary product, yields chitosan as a co-product from waste mycelia, might boost and increase the diffusion of this technology.

**Author Contributions:** S.C. contributed to data collection and revised the final paper; C.R. contributed to data collection and edited the final paper; M.P. acquired funding and revised the final paper; A.D. conceptualized the idea and wrote the original draft. All authors have read and agreed to the published version of the manuscript.

**Funding:** This study has received financial support from the Ministry of Education, University and Research (MIUR)—Italy, in the frame of the project PON 2015–2020: "ARS01_00985 BIOFEEDSTOCK—Development of Integrated Technological Platforms for Residual Biomass Exploitation".

**Institutional Review Board Statement:** Not applicable.

**Informed Consent Statement:** Not applicable.

**Data Availability Statement:** Not applicable.

**Conflicts of Interest:** The authors declare no conflict of interest.

## Abbreviations

AEAP, aqueous extract of apple pomace; AIM, alkali-insoluble material; CDA, chitin deacetylase; CS, chitin synthase; CSL, corn steep liquor; CVP, chitosan volumetric production; CWW, cassava wastewater; DD, degree of deacetylation; DW, deproteinized whey; F-6-P, fructose-6-phosphate; GA3, gibberellic acid; GlcN, D-glucosamine; GPY, glucose-peptone yeast extract; GYT, glucose-yeast extract-tryptone; GlcNAc, *N*-acetylglucosamine; GlcNAc-6-P, *N*-acetyl-D-glucosamine-6-phosphate; GlcNAc-1-P, *N*-acetyl-D-glucosamine-1-phosphate; LsF, liquid submerged fermentation; Mn, number average molecular weight; MW, molecular weight; PDB, potato dextrose broth; $P_i$, inorganic phosphorus; $r_P$, average hourly volumetric productivity; SSF, solid-state fermentation; UDP-GlcNAc, uridine 5′-diphospho-N-acetylglucosamine; YE, yeast extract; STR, stirred tank reactor; X, biomass volumetric production; YPD, yeast extract-peptone-dextrose.

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
