# Peer review of "Chitosan Production by Fungi: Current State of Knowledge, Future Opportunities and Constraints"

_fermentation, doi:10.3390/fermentation8020076_

Round 1
Reviewer 1 Report
The authors submitted a review article on chitosan, a polysaccharide, formed as a result of partial deacetylation of chitin. Shells of shrimp and other crustaceans are well-known sources of chitin, but the authors devoted their review article to fungal producers of chitin, and thus chitosan. The article is well-organized. Sufficient background and appropriate data were provided. Described materials are presented and summarized also using tables. In my opinion, the manuscript needs only some improvements.
I suggest adding chemical structures of chitin (especially with differentiation between α-, β- and γ-chitin) and chitosan.
Please take care of correctness in the chemical names, e.g. N-acetyl-D-glucosamine - "N" should be written in italics.
Line 33 - maybe there is a newer version of "Chitin and chitosan derivatives market report"
Line 78 - pKa instead of pka
Author Response
Referee 1
The authors submitted a review article on chitosan, a polysaccharide, formed as a result of partial deacetylation of chitin. Shells of shrimp and other crustaceans are well-known sources of chitin, but the authors devoted their review article to fungal producers of chitin, and thus chitosan. The article is well-organized. Sufficient background and appropriate data were provided. Described materials are presented and summarized also using tables. In my opinion, the manuscript needs only some improvements.
I suggest adding chemical structures of chitin (especially with differentiation between α-, β- and γ-chitin) and chitosan.
Answer: Accepted and a Figure (now Figure 1) added to the Ms.
Please take care of correctness in the chemical names, e.g. N-acetyl-D-glucosamine - "N" should be written in italics.
Answer: Accepted and all chemical names modified as requested
Line 33 - maybe there is a newer version of "Chitin and chitosan derivatives market report"
Answer: Accepted and the novel Report covering the 2020-2027 period by the same market agency added.
Line 78 - pKa instead of pka
Answer: Accepted and modified

Reviewer 2 Report
This is a relevant study on fungi producing chitin and chitosan, stressing the obvious need to use only GRAS species.
A good description of chitin and chitosan chemical properties and sources is made, as well as the production processes currently available, analyzing liquid and solid-state fermentation for fungal biomass. Then the factor affecting chitin and chitosan production are addressed, namely the factors that can be manipulated to increase biomass yield, as well as chitin and chitosan production. The last section of the review analyzes the concept of co-production of chitosan using mycelial wastes within a biorefinery concept, presenting some interesting solutions.
The review shows that fungal wastes over arthropod sources may be an interesting source of biomass to produce chitosan.
Yet, I would like to have seen some economical perspective of the final cost of such purified chitosan. Although the biomass may be cheap, the production, extraction, purification processes and environmental costs, are expensive and must be considered when deciding the best possible option.
This is, however, a minor issue regarding the review.
Overall, the document is very well written, clear, complete, and easy to understand. It is also duly structured in chapters and subchapters, which are organized in a logical and complete way.
The analysis of the scientific literature is comprehensive, the authors have consulted numerous relevant, and current articles.
Thus, I recommend publishing the paper with minor revision, enumerated below:
The chemical formulas of chitin and chitosan should be included
Line 80 – the only known cationic polymer so far, right?
Line 208, 460 – Zygomycetes with a capital letter
Line 258 – the maximum yield achieved was 32.4 g/kg for Mucor rouxii, grown in soybean meal substrate. According to this, you may need to revise the following sentence (lines 263-267).
Line 288 - although I recognize the work of Professor Stevens, I would not single out his title, to the exclusion of the other authors, also of merit.
Line 312 – format tables 3 and 4
Line 317 – rp in italic
Line 420 – yet, the liquid solutions seem to be less efficient than the solid solutions, don’t they? This should be discussed here as a final remark
Line 421 – it is section 3.4 (and following), not 3.3.
Line 463, 465 – in vivo, in vitro – in italic
Author Response
Referee 2
This is a relevant study on fungi producing chitin and chitosan, stressing the obvious need to use only GRAS species. A good description of chitin and chitosan chemical properties and sources is made, as well as the production processes currently available, analyzing liquid and solid-state fermentation for fungal biomass. Then the factor affecting chitin and chitosan production are addressed, namely the factors that can be manipulated to increase biomass yield, as well as chitin and chitosan production. The last section of the review analyzes the concept of co-production of chitosan using mycelial wastes within a biorefinery concept, presenting some interesting solutions. The review shows that fungal wastes over arthropod sources may be an interesting source of biomass to produce chitosan. Yet, I would like to have seen some economical perspective of the final cost of such purified chitosan. Although the biomass may be cheap, the production, extraction, purification processes and environmental costs, are expensive and must be considered when deciding the best possible option. This is, however, a minor issue regarding the review.
Overall, the document is very well written, clear, complete, and easy to understand. It is also duly structured in chapters and subchapters, which are organized in a logical and complete way. The analysis of the scientific literature is comprehensive, the authors have consulted numerous relevant, and current articles. Thus, I recommend publishing the paper with minor revision, enumerated below:
The chemical formulas of chitin and chitosan should be included
Answer: Accepted and a Figure (now Figure 1) added to the Ms.
Line 80 – the only known cationic polymer so far, right?
Answer: the term polymer has been replaced with glycan
Line 208, 460 – Zygomycetes with a capital letter
Answer: Accepted and modified
Line 258 – the maximum yield achieved was 32.4 g/kg for Mucor rouxii, grown in soybean meal substrate. According to this, you may need to revise the following sentence (lines 263-267).
Answer: Data shown at those lines refer to rP values. Thus, we have not revised that sentence
Line 288 - although I recognize the work of Professor Stevens, I would not single out his title, to the exclusion of the other authors, also of merit.
Answer: Accepted and modified as requested.
Line 312 – format tables 3 and 4
Answer: Tables have been formatted
Line 317 – rp in italic
Answer: Accepted and modified at li. 310
Line 420 – yet, the liquid solutions seem to be less efficient than the solid solutions, don’t they? This should be discussed here as a final remark
Answer: if we have not misinterpreted the Reviewer’s comment, it is an invitation to compare the production performance of SSF and LSF. Unfortunately, there are only few studies that focused on this comparison. Some of them pointed out the superiority of SSF over LSF [104, 109] and another one led to opposite outcome [60]. We have added other considerations related to this comparison in the Conclusions section.
Line 421 – it is section 3.4 (and following), not 3.3.
Answer: Accepted and modified
Line 463, 465 – in vivo, in vitro – in italic
Answer: Accepted and modified here and everywhere
